# Conformal maps and superfluid vortex dynamics on curved and bounded surfaces: The case of an elliptical boundary

Matteo Caldara[1⋆], Andrea Richaud[2], Pietro Massignan[2†] and Alexander L. Fetter[3]

**1** Scuola Internazionale Superiore di Studi Avanzati (SISSA),
Via Bonomea 265, I-34136, Trieste, Italy
**2** Departament de Física, Universitat Politècnica de Catalunya,
Campus Nord B4-B5, E-08034 Barcelona, Spain
**3** Department of Physics and Department of Applied Physics,
Stanford University, Stanford, California 94305-4045, USA

⋆ mcaldara@sissa.it , † pietro.massignan@upc.edu

## Abstract

Recent advances in cold-atom platforms have made real-time dynamics accessible, renewing interest in the motion of superfluid vortices in two-dimensional domains. Here we show that the energy and the trajectories of arbitrary vortex configurations may be computed on a complicated (curved or bounded) surface, provided that one knows a conformal map that links the latter to a simpler domain (like the full plane, or a circular boundary). We also prove that Hamilton's equations based on the vortex energy agree with the complex dynamical equations for the vortex dynamics, demonstrating that the vortex trajectories are constant-energy curves. We use these ideas to study the dynamics of vortices in a two-dimensional incompressible superfluid with an elliptical boundary, and we derive an analytical expression for the complex potential describing the hydrodynamic flow throughout the fluid. For a vortex inside an elliptical boundary, the orbits are nearly self-similar ellipses.



## Contents



# 1 Introduction

Quantized vortices are fundamental topological excitations of superfluids, such as liquid $^4$He-II [1], dilute one- and two-component Bose-Einstein condensates (BECs), and two-component fermionic mixtures [2, 3]. At low temperatures, these systems become nearly ideal fluids with negligible viscosity and dissipation. As a result, superfluid vortices obey the dynamical equations of classical hydrodynamics [4, 5] augmented with the condition of quantized vorticity [6, 7].

Three-dimensional vortex lines have various bending modes that complicate the analysis of their dynamics. The situation becomes much simpler in two dimensions because only translational motion remains. In addition, for dilute-gas BECs, the typical diameter of the vortex cores is much less than all other length scales like the trap size or intervortex separation. Hence these vortices act like point vortices, with the $x$ and $y$ coordinates as canonically conjugate variables and first-order equations of motion.

In the 1960s, experiments with rotating superfluid He-II [8, 9] stimulated theoretical studies of equilibrium two-dimensional (2D) vortex states in cylinders [10] and annuli [11]. More recently, some of us have studied the dynamics of quantized vortices on nonplanar 2D surfaces [12–15] relying on a complex potential that exploits the properties of conformal transformations.

It is useful to describe a superfluid system at low temperature with a macroscopic condensate wave function $\Psi(\boldsymbol{r}) = \sqrt{n(\boldsymbol{r})}e^{i\Phi(\boldsymbol{r})}$ in terms of two real fields, the number density $n(\boldsymbol{r})$ and the phase $\Phi(\boldsymbol{r})$. The latter determines the two-dimensional superfluid velocity through $\boldsymbol{v} = \hbar\boldsymbol{\nabla}\Phi/M$, with $M$ the atomic mass. The flow is then irrotational $\boldsymbol{\nabla} \times \boldsymbol{v} = 0$ everywhere except at the phase singularities associated with the vortex cores.

Many cold-atom experimental platforms are able to produce essentially uniform systems [16, 17], with negligible local changes of the density in the bulk of the superfluid. In the absence of sound waves from acoustic excitations, the continuity equation for particle conservation, $\partial_t n + \boldsymbol{\nabla} \cdot (n\boldsymbol{v}) = 0$, implies that the flow is effectively incompressible, with $\boldsymbol{\nabla} \cdot \boldsymbol{v} = 0$.

As a consequence, such a two-dimensional flow may also be described by the stream function $\chi(\boldsymbol{r})$, in terms of which the superfluid velocity in the $xy$ plane becomes $\boldsymbol{v} = (\hbar/M)\hat{\boldsymbol{z}} \times \boldsymbol{\nabla}\chi$.

We now have two distinct representations of the hydrodynamic velocity $\boldsymbol{v}$. When written out in detail, the cartesian components of $\boldsymbol{v}$ satisfy the Cauchy-Riemann equations. Hence $\chi$ and $\Phi$ can be interpreted as the real and imaginary parts of an analytic function of a complex variable $z = x + iy$. In this way, we construct the complex potential defined as

$$F(z) = \chi(\boldsymbol{r}) + i\Phi(\boldsymbol{r}), \tag{1}$$

with $\boldsymbol{r}$ the two-dimensional position vector. The Cauchy-Riemann conditions give the following compact representation of the hydrodynamic flow velocity:

$$v_y + iv_x = \frac{\hbar}{M}\frac{dF(z)}{dz}. \tag{2}$$

Early experiments on rotating $^4$He-II used circular containers with rotationally invariant walls. For such a geometry, surface roughness at the wall triggers the nucleation of vortices, which then migrate into the bulk of the superfluid. In contrast, a rotating elliptical boundary pushes the superfluid, imparting angular momentum even though the flow remains irrotational for slow rotations. As the rotation rate increases, however, isolated vortices eventually appear within the container [18]. These predictions found a solid confirmation in experimental studies of vortex states in rotating superfluid $^4$He-II for three elliptical containers with different eccentricities [19].

Reference [18] focused on the energy and angular momentum of a vortex in an elliptical boundary, with no consideration of the associated vortex dynamics, which was experimentally inaccessible at that time. More recently, the creation of cold-atom BECs has allowed direct real-time studies of vortex dynamics [20–22]. In this context, we study here the motion of two-dimensional superfluid vortices outside and inside a stationary elliptical boundary. An additional motivation is the recent experimental accessibility of such configurations using digital micromirror devices (DMDs) [17, 23–27].

Previous studies of vortices [18] or two-dimensional point charges [28] with elliptical boundaries have used standard methods of mathematical physics with elliptic coordinates [29], leading to real solutions expressed as infinite series whose convergence requires detailed analysis. Here, instead, we rely on complex variables and conformal maps to solve the same problems, giving explicit solutions expressed in terms of well-known functions of mathematical physics.

The recent study [28] of two-dimensional point charges outside and inside an elliptical boundary obtained infinite series for the electrostatic potential and focused on finding equivalent sets of charged images. The similar problem of point vortices with the same geometry is far richer. In addition to determining the stream function $\chi$ for the vortices (analogous to the electrostatic potential for the charges), the dynamics of the vortices is also of great interest, particularly because of experiments with dilute-gas BECs. In addition, our complex formalism also gives the phase pattern $\Phi$ of the vortices (analogous to the electric field lines).

Reference [30] used a conformal transformation to relate the energy of a vortex on a curved surface to the corresponding energy on a simpler surface through the metric properties of the conformal map relating the two surfaces. Section 2 reviews and extends this technique to obtain the complex dynamics of a single vortex inside a general closed boundary in terms of the complex dynamics in the simpler geometry. We also show that these dynamical equations agree with the real Hamilton's equations based on the energy as a function of the coordinates of the vortex. In Sec. 3, we study a single vortex both inside and outside a circular boundary, which serves as our simple geometry. Section 4 presents the Joukowsky transformation and its analytic properties that play an important role in our analysis. In Sec. 5 we study a single

superfluid vortex outside a two-dimensional elliptical boundary. In the subsequent Sec. 6 we solve the more intricate problem of a single vortex in an elliptical domain. We obtain an explicit analytical expression for the complex potential, leading to the equations of motion for a vortex inside an elliptical boundary. Their numerical solution gives the vortex trajectories, which are approximately but not exactly elliptical. We also generalize to a multivortex configuration, showing results for a symmetric vortex dipole. Finally, in Sec. 7, we summarize and suggest possible future extensions of our work. Appendix A presents an alternative (but equivalent) treatment of a vortex inside an ellipse, based on a direct transformation from a circle to the ellipse.

## 2 Conformal maps and basic physical properties

We here study the behaviour of a single positive quantized vortex either inside or outside a quite general curved boundary. It is convenient to solve this problem with a conformal map from a simpler standard boundary (for example a circle) to the general boundary (for example an ellipse). Let $z$ be the complex plane with the standard boundary and $w$ be the complex plane with the general boundary. Assume that $w(z)$ is the conformal map from the standard boundary to the general boundary, with inverse $z(w)$. Infinitesimal displacements on the two surfaces are related by

$$dz = e^{\sigma(w)}dw, \tag{3}$$

which defines the space-dependent complex scale factor

$$\frac{dz(w)}{dw} = z'(w) = e^{\sigma(w)} \quad \text{or, equivalently,} \quad \sigma(w) = \ln\big(z'(w)\big). \tag{4}$$

Reference [30] studies a real infinitesimal displacement with a real scale factor $e^{\omega(w)}$. They are essentially the same, with $\omega(w) = \text{Re}\,\sigma(w)$. In the $z$ plane, a vortex at $z_0$ has the complex potential $F_z(z; z_0)$ that is known explicitly. The corresponding complex potential for the general boundary in the $w$ plane is

$$F_w(w; w_0) = F_z[z(w); z(w_0)]. \tag{5}$$

### 2.1 Energy of a single vortex

Since the energy $E_z = \frac{1}{2}nM \int d^2r\,|\boldsymbol{v}|^2$ in the $z$ plane is purely kinetic, it is simple to find $E_z$ of a single vortex using the stream function $\chi = \text{Re}\,F_z(z; z_0)$ and the relation for the flow velocity $\boldsymbol{v} = (\hbar/M)\hat{\boldsymbol{n}} \times \boldsymbol{\nabla}\chi$, where $\hat{\boldsymbol{n}}$ is the unit normal to the $z$ plane. Specifically, $E_z = \frac{1}{2}\hbar n \int d^2r\,\boldsymbol{v}\cdot\hat{\boldsymbol{n}}\times\boldsymbol{\nabla}\chi$. Some vector manipulations lead to

$$E_z = \frac{\hbar n}{2}\int d^2r\,[-\boldsymbol{\nabla}\cdot(\chi\hat{\boldsymbol{n}}\times\boldsymbol{v}) - \chi\hat{\boldsymbol{n}}\cdot(\boldsymbol{\nabla}\times\boldsymbol{v})]. \tag{6}$$

In the first term, the divergence theorem gives integrals of the form $\oint d\boldsymbol{l}\cdot\boldsymbol{v}\chi$ around each boundary. The stream function on the $j^{\text{th}}$ boundary takes the constant value $\chi_j$ and the remaining line integral is a positive or negative integer $\mathfrak{n}_j$ times $2\pi\hbar/M$, where the line integral is in the positive sense and the sign of $\mathfrak{n}_j$ depends on the sense of the flow around the boundary.

The second term involves the vorticity $\boldsymbol{\nabla}\times\boldsymbol{v} = (2\pi\hbar\hat{\boldsymbol{n}}/M)\delta^{(2)}(\boldsymbol{r} - \boldsymbol{r}_0)$, centered at the vortex position, where the stream function diverges logarithmically. The integral thus reduces

to a sum of two terms evaluated inside the vortex core (0-subscript):

$$\int d^2 r\, \chi \hat{\boldsymbol{n}} \cdot (\boldsymbol{\nabla} \times \boldsymbol{v}) = \int_0 d^2 r\, \hat{\boldsymbol{n}} \cdot (\boldsymbol{\nabla} \times \boldsymbol{v})(\chi - \ln|\boldsymbol{r} - \boldsymbol{r}_0|) + \int_0 d^2 r\, \hat{\boldsymbol{n}} \cdot (\boldsymbol{\nabla} \times \boldsymbol{v}) \ln|\boldsymbol{r} - \boldsymbol{r}_0|$$
$$= \frac{2\pi\hbar}{M} \lim_{r \to r_0} [\chi(\boldsymbol{r}) - \ln|\boldsymbol{r} - \boldsymbol{r}_0|]\,, \tag{7}$$

where we assume that the vortex has a small hollow core excluding the vorticity at its center. A combination of these results gives

$$E_z = \frac{\pi\hbar^2 n}{M}\left(\sum_j \mathfrak{n}_j \chi_j - \tilde{\chi}_0^z\right)\,, \tag{8}$$

where $\tilde{\chi}_0^z = \mathrm{Re}\lim_{z \to z_0}[F_z(z; z_0) - \ln(z - z_0)]$ is the regularized stream function. A completely analogous formula holds for the energy in the $w$ plane $E_w = (\pi\hbar^2 n/M)\left(\sum_j \mathfrak{n}_j \chi_j - \tilde{\chi}_0^w\right)$.

Now consider the difference $E_w - E_z$. The conformal transformation conserves both the quantization integers $\mathfrak{n}_j$ and the circulation constants $\chi_j$, so that

$$E_w - E_z = \frac{\pi\hbar^2 n}{M}\left[\tilde{\chi}_0^z(z(w_0)) - \tilde{\chi}_0^w(w_0)\right]$$
$$= -\frac{\pi\hbar^2 n}{M}\mathrm{Re}\lim_{w \to w_0} \ln\left[\frac{z(w) - z(w_0)}{w - w_0}\right]\,.$$

The second line is obtained using the definitions of the regularized stream functions, together with the properties of the conformal map which ensure that the terms involving the full complex potential cancel. Writing $w = w_0 + \delta$ and expanding $z(w) \approx z(w_0) + \delta z'(w_0)$ gives the very simple and general result

$$E_w = E_z - \frac{\pi\hbar^2 n}{M}\mathrm{Re}\,\sigma(w_0)\,, \tag{9}$$

where $\sigma(w)$ is the scale factor defined in Eq. (4).

## 2.2 Dynamics of a single vortex

For a general complex potential $F(z)$, Eq. (2) gives the hydrodynamic velocity including the circulating flow around the vortex itself. This latter flow does not affect the vortex dynamics and must be subtracted off. In this way, the vortex in the $z$ plane obeys the dynamical equation

$$i\dot{z}_0^* = \frac{\hbar}{M}\left[\frac{dF_z(z; z_0)}{dz} - \frac{1}{z - z_0}\right]_{z \to z_0}\,, \tag{10}$$

where $^*$ denotes complex conjugation. Similarly, the vortex at $w_0$ in the $w$ plane has the dynamical equation

$$i\dot{w}_0^* = \frac{\hbar}{M}\left[\frac{dF_w(w; w_0)}{dw} - \frac{1}{w - w_0}\right]_{w \to w_0}\,. \tag{11}$$

Here, however, we have a conformal map $w(z)$ relating the two planes, so that the complex potential in the $w$ plane becomes $F_w(w; w_0) = F_z[z(w); z(w_0)]$. As a result, we can write

$$\frac{dF_w(w; w_0)}{dw} = z'(w)\frac{dF_z(z(w); z(w_0))}{dz}\,,$$

giving

$$i\dot{w}_0^* = z'(w_0)\,i\dot{z}_0^* + \frac{\hbar}{M}\left[\frac{z'(w)}{z(w)-z(w_0)} - \frac{1}{w-w_0}\right]_{w\to w_0},\tag{12}$$

where $\dot{z}_0^*$ has to be understood as a function of $w_0$. We again write $w = w_0 + \delta$ and expand $z(w)-z(w_0) \approx \delta z'(w_0) + \frac{1}{2}\delta^2 z''(w_0)$. The final result becomes

$$i\dot{w}_0^* = i\dot{z}_0^* e^{\sigma(w_0)} + \frac{\hbar}{2M}\sigma'(w_0),\tag{13}$$

where $\sigma'(w_0) = (d\sigma(w)/dw)_{w_0}$. Integration of this complex equation would give the time-dependent vortex dynamics $\{x_0(t), y_0(t)\}$.

### 2.2.1 Verification of Hamilton's equations

Hamilton's equations for vortex dynamics with a closed boundary have the familiar form

$$\dot{x}_0 = \frac{\partial E_w/(2\pi\hbar n)}{\partial y_0}, \qquad \dot{y}_0 = -\frac{\partial E_w/(2\pi\hbar n)}{\partial x_0},\tag{14}$$

involving $E_w(x_0, y_0)$. Here, we verify that they are equivalent to the complex dynamics in Eq. (13).

The stream function is the real part of the complex potential, so that the general expression (8) for the energy has the form $E_w = (\pi\hbar^2 n/M)\,\mathrm{Re}\,\mathcal{E}_w(w_0, w_0^*)$. In general, $\mathcal{E}_w(w_0, w_0^*)$ is a complex function of both $w_0$ and $w_0^*$. With $w_0 = x_0 + iy_0$ and $w_0^* = x_0 - iy_0$, Hamilton's equations become

$$\frac{2M\dot{y}_0}{\hbar} = -\mathrm{Re}\left(\frac{\partial\mathcal{E}_w}{\partial w_0} + \frac{\partial\mathcal{E}_w}{\partial w_0^*}\right) \quad\text{and}\quad \frac{2M\dot{x}_0}{\hbar} = -\mathrm{Im}\left(\frac{\partial\mathcal{E}_w}{\partial w_0} - \frac{\partial\mathcal{E}_w}{\partial w_0^*}\right),$$

from which the basic result immediately follows:

$$\dot{y}_0 + i\dot{x}_0 = i\dot{w}_0^* = -\frac{\hbar}{2M}\left[\frac{\partial\mathcal{E}_w}{\partial w_0} + \left(\frac{\partial\mathcal{E}_w}{\partial w_0^*}\right)^*\right].\tag{15}$$

Equation (9) shows that $\mathcal{E}_w(w_0, w_0^*) = \mathcal{E}_z[z(w_0), z^*(w_0^*)] - \sigma(w_0)$, where $\mathcal{E}_z(z, z^*)$ follows from the real part of the complex potential $F_z(z; z_0)$. Hence Eq. (15) becomes

$$\begin{aligned}i\dot{w}_0^* &= -\frac{\hbar}{2M}\left[\left.\frac{\partial\mathcal{E}_z(z,z^*)}{\partial z}\right|_{z(w_0)}\frac{dz(w_0)}{dw_0} + \left(\left.\frac{\partial\mathcal{E}_z(z,z^*)}{\partial z^*}\right|_{z^*(w_0^*)}\frac{dz^*(w_0^*)}{dw_0^*}\right)^* - \sigma'(w_0)\right]\\&= i\dot{z}_0^* e^{\sigma(w_0)} + \frac{\hbar}{2M}\sigma'(w_0).\end{aligned}\tag{16}$$

This complex dynamical equation is the same as Eq. (13). Since Hamilton's equations conserve the energy, the equivalence with Hamilton's equations ensures that the vortex orbits are closed and the same as the closed curves of constant energy.

## 2.3 Generalization to configurations with multiple vortices

For a collection of $N_v$ vortices with charges $c_j$ located at positions $w_j$ the vorticity reads $\boldsymbol{\nabla}\times\boldsymbol{v} = (2\pi\hbar\hat{\boldsymbol{n}}/M)\sum_{j=1}^{N_v} c_j\delta^{(2)}(\boldsymbol{r}-\boldsymbol{r}_j)$, and the above relations readily generalize to

$$E_w = E_z - \frac{\pi\hbar^2 n}{M}\sum_{j=1}^{N_v} c_j^2\,\mathrm{Re}\,\sigma(w_j),\tag{17}$$

and

$$iw_j^* = i\dot{z}_j^* e^{\sigma(w_j)} + \frac{\hbar c_j}{2M}\sigma'(w_j).\tag{18}$$

Reference [30] obtained our Eq. (17) in their Eq. (85), but the approach outlined above seems simpler and more direct. In contrast, our relation (18) for the vortex velocity is new, since Ref. [30] explicitly excluded vortex dynamics from its consideration. Interestingly, the contributions to the energy arising from the scale factors are independent of the sign of the vortices.

## 2.4 Example: Conformal map from a plane to the surface of a cylinder

The conformal map $z_\pm(w) = e^{\pm iw}$ takes the flat $z$ plane to the curved $w$ surface of a cylinder with unit radius. We write $w = x + iy$ so that $z_\pm(w) = e^{\pm ix}\,e^{\mp y}$. This transformation is $2\pi$ periodic in $x$, which is the angular direction around the cylinder, while $y$ becomes the axial direction along the length of the cylinder. Appendix A of Ref. [12] discusses this transformation in more detail. Here, we note that $z_\pm'(w_0) = e^{\sigma^\pm(w_0)} = \pm ie^{\pm iw_0}$, so that $\sigma^\pm(w_0) = \pm i\pi/2 \pm iw_0$.

Since a single vortex in an unbounded $z$ plane remains stationary, the term $i\dot{z}_0^*$ in Eq. (13) vanishes, leaving the result $i\dot{w}_0^* = i\dot{x}_0 + \dot{y}_0 = \pm i\hbar/(2M)$. This means that the vortex precesses uniformly around the cylinder with quantized speed $\pm\hbar/(2M)$ in either direction at fixed height $y_0$, as found in Ref. [12] by a different method. This example is interesting because the motion of the vortex arises solely from the scale factor.

Similarly, the interaction energy of a vortex dipole at $z_1$ and $z_2$ on the $z$ plane is $E_z = (2\pi\hbar^2 n/M)\,\mathrm{Re}\ln(z_1 - z_2)$. The interaction energy of a dipole with vortices at $w_1$ and $w_2$ on the surface of a cylinder is readily found using Eq. (17) to be

$$\begin{aligned}
E_w &= \frac{2\pi\hbar^2 n}{M}\left\{\mathrm{Re}\ln\left[z_+(w_1) - z_+(w_2)\right] - \frac{1}{2}\mathrm{Re}\left[\sigma^+(w_1) + \sigma^+(w_2)\right]\right\}\\
&= \frac{2\pi\hbar^2 n}{M}\,\mathrm{Re}\ln\left[\sin\left(\frac{w_1 - w_2}{2}\right)\right] + \mathrm{const}.
\end{aligned}\tag{19}$$

This result coincides with the one found via a more complicated route in Eq. (27) of Ref. [12].

# 3 Single vortex with a circular boundary

We now apply this general formalism, with the $z$ plane containing a circular boundary of radius $R$ and the $w$ plane containing an elliptical boundary. In this section, we review the elementary solutions for a single vortex at $z_0$ both inside and outside a circular boundary of radius $R$ [5]. They rely on a single opposite-sign image vortex at $z_0' = R^2/z_0^*$ on the opposite side of the circular boundary. The following section then describes the Joukowsky conformal transformation that maps concentric circles in the $z$ plane to confocal ellipses in the $w$ plane.

## 3.1 Vortex dynamics

As noted above, the complex potential for a vortex with a circular boundary involves a single opposite-sign image. We now present the two cases of a vortex inside/outside the circular boundary, which require an image vortex outside/inside the boundary.

### 3.1.1 Vortex inside circular boundary

The complex potential for a positive vortex on the $z$ plane inside a circular boundary of radius $R$ is

$$F_{\mathrm{inside-circle}}(z;z_0) = \ln(z - z_0) - \ln(z - z_0').\tag{20}$$

The first term of Eq. (20) describes circulating flow around the vortex and does not contribute to the vortex dynamics. As a result, only the second term is relevant and we have the complex dynamical equation

$$i\dot{z}_0^* = \frac{\hbar}{Mz_0} \frac{|z_0|^2}{R^2 - |z_0|^2}. \tag{21}$$

For this interior vortex the uniform precession rate is

$$\dot{\theta}_0 = \frac{\hbar}{M(R^2 - r_0^2)}. \tag{22}$$

### 3.1.2  Vortex outside circular boundary

The complex potential for a positive vortex on the $z$ plane outside the circular boundary is

$$F_{\text{outside-circle}}(z; z_0) = \ln(z - z_0) - \ln\left(z - z_0'\right) + \mathfrak{n}_0 \ln z. \tag{23}$$

Here the last term represents an $\mathfrak{n}_0$-fold quantized circulation around the origin, where $\mathfrak{n}_0$ is a general integer. Note that the circulation $\mathfrak{n}_i$ around the circular boundary at radius $R$ includes both the central vortex and the negative image at distance $R^2/r_0 < R$, giving $\mathfrak{n}_i = \mathfrak{n}_0 - 1$, using the general notation from Sec. 2.1. It is straightforward to find the precession rate for a vortex outside the circular boundary, that is

$$\dot{\theta}_0 = \frac{\hbar}{Mr_0^2}\left(\mathfrak{n}_0 - \frac{r_0^2}{r_0^2 - R^2}\right) = \frac{\hbar}{Mr_0^2}\left(\mathfrak{n}_i - \frac{R^2}{r_0^2 - R^2}\right). \tag{24}$$

This result reproduces Eq. (B8) in Ref. [12].

## 3.2  Energy of one vortex

We now use Eq. (8) to find the energy $E_z$ of a single vortex with a circular boundary. Specifically, we need the stream function

$$\chi_{\text{circle}}(\mathbf{r}; \mathbf{r}_0) = \text{Re}\, F_{\text{circle}}(z; z_0) = \mathfrak{n}_0 \ln r + \ln|\mathbf{r} - \mathbf{r}_0| - \ln|\mathbf{r} - \mathbf{r}_0'|, \tag{25}$$

with $\mathbf{r} = (r, \theta)$ in polar coordinates and similarly for $\mathbf{r}_0 = (r_0, \theta_0)$ and $\mathbf{r}_0' = (R^2/r_0, \theta_0)$.

For an exterior vortex, there are two boundaries: an outer circle at $R_\infty$ where $\mathfrak{n}_{\text{out}} = \mathfrak{n}_0$ and $\chi_{\text{out}} = \mathfrak{n}_0 \ln R_\infty$; and an inner circle at $R$ where $\mathfrak{n}_{\text{in}} = -\mathfrak{n}_0 + 1$ and $\chi_{\text{in}} = \mathfrak{n}_0 \ln R + \ln(r_0/R)$. In addition, we need $\tilde{\chi}_0^z = \mathfrak{n}_0 \ln r_0 - \ln|r_0 - R^2/r_0|$. Substitution into Eq. (8) gives the energy

$$E_{\text{circle}} = \frac{\pi\hbar^2 n}{M}\left[\mathfrak{n}_0^2 \ln\left(\frac{R_\infty}{R}\right) - 2\mathfrak{n}_0 \ln\left(\frac{r_0}{R}\right) - \ln R + \ln\left|r_0^2 - R^2\right|\right]. \tag{26}$$

As a check on this expression, $E_{\text{circle}}$ depends only on $r_0$. Conservation of energy and Hamilton's equations confirm that the vortex moves on closed circular orbits with angular velocity (22) or (24).

## 4  Joukowsky map

The Joukowsky map [31] usually appears in connection with airfoil design. Here, instead, we focus on its simpler property of mapping a family of concentric circles into a family of confocal ellipses. Specifically, we study the Joukowsky map from the $z$ plane to the $w = x + iy$ plane

$$w = \frac{1}{2}\left(z + z^{-1}\right). \tag{27}$$

Consider the circle $z = Re^{i\theta}$, with $R > 1$. The Joukowsky transformation maps this circle in the $z$ plane into the parametric curve $x = \frac{1}{2}(R + R^{-1})\cos\theta$ and $y = \frac{1}{2}(R - R^{-1})\sin\theta$ in the $w$ plane. Eliminating $\theta$ gives the ellipse $x^2/a^2 + y^2/b^2 = 1$ with semimajor axis $a = \frac{1}{2}(R + R^{-1})$ and semiminor axis $b = \frac{1}{2}(R - R^{-1})$. The inverse relations are $R = a + b$ and $R^{-1} = a - b$. In addition we have $a^2 - b^2 = 1$ so that the ellipse has focal points at $w = \pm 1$.

Note that the limit $R \to \infty$ yields a large circle while the limit $R \to 1$ yields a flat ellipse encircling the focal line. This discussion holds for any $R$, so that the Joukowsky transformation maps a family of concentric circles in the $z$ plane into a family of confocal ellipses in the $w$ plane.

The Joukowsky transformation has several interesting properties:

1. The function (27) has a simple pole at $z = 0$. The differential $dw = w'(z)dz$ gives the element of squared length $|dw|^2 = |w'(z)|^2|dz|^2$. The transformation is conformal everywhere except at $z = \pm 1$ where $w'(z) = \frac{1}{2}(1 - z^{-2})$ vanishes. These points map to the focal points at $w = \pm 1$.

2. The Joukowsky transformation is symmetric under the interchange $z \leftrightarrow z^{-1}$. This inversion symmetry means that the outer circle at $R$ and the inner circle at the inversion radius $R^{-1}$ form the boundaries of an annulus. In addition, two points in the $z$ plane, one outside the unit circle and the other inside the unit circle, both map into the same point in the $w$ plane.

3. It will be important to consider the inverse function $z(w)$, and it is straightforward to find the two roots $z_\pm(w) = w \pm \sqrt{w^2 - 1}$. As expected from the inversion symmetry, we have $z_+(w)z_-(w) = 1$. We choose to put a branch cut between the focal points at $w = \pm 1$ with the function $z_+(w)$ real and positive for $w \to +\infty$ along the positive real axis. The two roots $z_\pm(w)$ form two Riemann sheets; each region $|z| < 1$ and $|z| > 1$ maps onto the whole $w$ plane.

## 5 Vortex outside an elliptical boundary

It is now easy to use the results of Sec. 2 to find the energy and dynamics of a single vortex outside an elliptical boundary by combining the Joukowsky transformation with the corresponding results for a vortex outside a circular boundary. Since the vortex is outside, the relevant root has the $+$ sign, and we use the notation

$$\mathfrak{z}(w) = w + \sqrt{w^2 - 1}. \tag{28}$$

Here the procedure is straightforward because the branch cut of the inverse function $\mathfrak{z}(w)$ lies in the excluded interior region of the ellipse. As shown in the following Sec. 6, the situation for a vortex inside the elliptical boundary requires a more careful treatment because the branch cut lies inside the physical region.

### 5.1 Complex potential for a vortex outside an elliptical boundary

We start from the complex potential Eq. (23) and use the Joukowsky map to find

$$
\begin{aligned}
F_{\text{outside}-\text{ellipse}}(w; w_0) &= F_{\text{circle}}(\mathfrak{z}(w); \mathfrak{z}(w_0)) \\
&= \mathfrak{n}_0 \ln \mathfrak{z}(w) + \ln[\mathfrak{z}(w) - \mathfrak{z}(w_0)] - \ln\left[\mathfrak{z}(w) - \frac{(a+b)^2}{\mathfrak{z}(w_0)^*}\right]. \tag{29}
\end{aligned}
$$

The real part is the stream function and the imaginary part is the phase function. Figure 1 uses these real functions to plot the stream lines and phase pattern for two cases: $\mathfrak{n}_0 = 0$ and $\mathfrak{n}_0 = 1$. These figures have all the expected features and verify that the proposed Joukowsky mapping from a circle to an ellipse is correct.

Majic [28] used elliptic coordinates to study the potential of a two-dimensional point charge outside an elliptical grounded conducting surface. The resulting real expression was an infinite series, in contrast to the real part of the complex potential in Eq. (29), which is a simple sum of three logarithms. Moreover, our formalism has the advantage of giving not only the streamlines, but also the phase plots. As shown below, it yields many other physical results such as the vortex dynamics and the vortex self energy $E_{\text{outside−ellipse}}$, all expressed explicitly in terms of well-known mathematical functions.

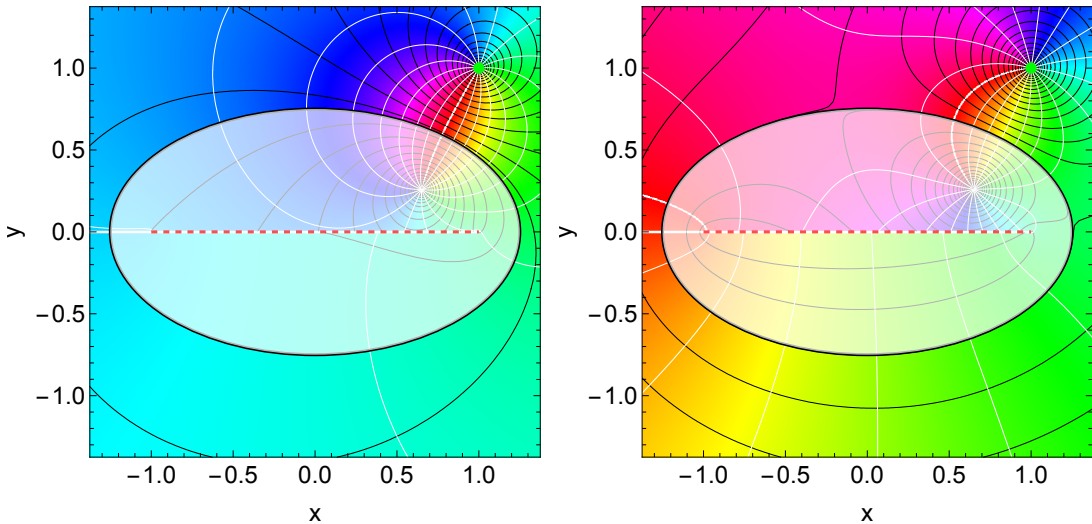

Figure 1: Real and imaginary parts of Eq. (29) giving streamlines (black lines) and phase (colour coding, and white lines) for a vortex outside of an ellipse (indicated respectively by the bright green dot and the black thick contour). The ellipse has aspect ratio $b/a = 0.6$, and its two foci are joined by the red dashed line. Left panel: $\mathfrak{n}_0 = 0$, meaning circulation $\mathfrak{n}_i = -1$ around the elliptical boundary. Right panel: $\mathfrak{n}_0 = 1$, meaning circulation $\mathfrak{n}_i = 0$ around the elliptical boundary.

## 5.2 Energy of a vortex outside an elliptical boundary

Equation (9) applies directly here, using Eq. (26) for $E_z$. For the circular boundary, the radial position is $r_0 = |z_0|$ with $R = a + b$ and $R_\infty = a_\infty$. The inverse transformation in Eq. (28) gives $d\mathfrak{z}(w)/dw = \mathfrak{z}(w)/\sqrt{w^2 - 1}$, so that $\sigma(w) = \ln \mathfrak{z}(w) - \ln(w^2 - 1)/2$. In this way, we find the compact result for the energy of a vortex at $w_0$ outside an elliptical boundary

$$
\begin{aligned}
E_{\text{outside−ellipse}} = \frac{\pi \hbar^2 n}{M} \Bigg[ & \mathfrak{n}_0^2 \ln\left(\frac{a_\infty}{a+b}\right) - 2\mathfrak{n}_0 \ln\left(\frac{|\mathfrak{z}(w_0)|}{a+b}\right) - \ln(a+b) \\
& + \ln\left(|\mathfrak{z}(w_0)|^2 - (a+b)^2\right) + \frac{1}{2}\ln|w_0^2 - 1| - \ln|\mathfrak{z}(w_0)| \Bigg].
\end{aligned}
\tag{30}
$$

Equipotential curves of this energy for $\mathfrak{n}_0 = 0$ are closed vortex orbits around the elliptical boundary, as shown in Fig. 2. To understand this figure, we consider a single vortex at a small distance $d \ll a$ outside the elliptical boundary. If $d$ is much smaller than the local radius of curvature, then the boundary is effectively flat. In this case, the condition of tangential flow at the boundary requires an opposite-sign image at a distance $d$ inside the boundary. The vortex

and its effective image dominate the complex potential, which then approximates that of a vortex dipole with separation $2d$. The energy has a logarithmic dependence $\propto \ln(2d)$, and the translational velocity is $\propto 1/(2d)$ [5]. Considering the simple case $w_0 = a + d$ and Taylor expanding up to linear order in $d/a \ll 1$, one can see that the logarithmic behaviour of the energy arises from the first term in the second line of Eq. (30). Conservation of energy requires that $d$ remains constant, so that the corresponding closed constant-energy curve follows the shape of the boundary. The increased density of contour lines near the boundary reflects the increasingly rapid precession near the elliptical surface, as expected from Hamilton's equations.

The situation is quite different for a vortex far from the boundary, as seen from Fig. 2, where such curves become circular. In the limit $|w_0| \gg 1$, in fact, $|\mathfrak{Z}(w_0)| \approx 2|w_0|$ and $\frac{1}{2}\ln|w_0^2 - 1| \approx \ln|w_0|$, so that the energy (30) reduces to Eq. (26) for a vortex outside a circular boundary.

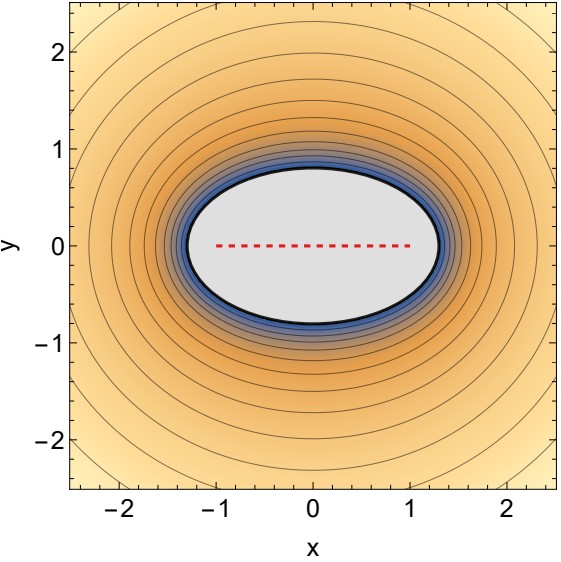

Figure 2: Contours of equal energy for a flow with $\mathfrak{n}_0 = 0$ around an ellipse with aspect ratio $b/a = 0.6$. The energy varies monotonically from large negative values (dark blue) to large positive ones (light orange).

## 5.3 Dynamics of a vortex outside an elliptical boundary

We now combine Eq. (21) for the dynamics of a single vortex outside a circle of radius $R$ with the general relation (13) to find the dynamical equation for a vortex outside an elliptical boundary with given semiaxes $a$ and $b$. Since

$$\sigma'(w) = -\frac{1}{\mathfrak{Z}(w)(w^2 - 1)} = \frac{1}{\sqrt{w^2 - 1}}\left(1 - \frac{w}{\sqrt{w^2 - 1}}\right),$$

we find

$$i\dot{w}_0^* = \frac{\hbar}{M\sqrt{w_0^2 - 1}}\left[\mathfrak{n}_0 - \frac{|\mathfrak{Z}(w_0)|^2}{|\mathfrak{Z}(w_0)|^2 - (a + b)^2} + \frac{1}{2}\left(1 - \frac{w_0}{\sqrt{w_0^2 - 1}}\right)\right]. \tag{31}$$

For this vortex outside an elliptical boundary, our work in Sec. 2.2.1 ensures that Hamilton's equations give the same vortex dynamics as here in Eq. (31). Hence the closed vortex orbits coincide with the contours of equal energy in Fig. 2.

# 6 Vortex inside an elliptical boundary

As seen previously, the complex potential for a single vortex at $z_0 = r_0 e^{i\theta_0}$ inside a circular boundary is slightly simpler than that for a single vortex at $z_0$ outside the same circular boundary, but only because the external position allows the possibility of additional $\mathfrak{n}_0$-fold central circulation. For both inside and outside positions, a single image at $z_0' = r_0' e^{i\theta_0}$ is required, where $r_0' = R^2/r_0$. Since $r_0 r_0' = R^2$, the original vortex and its image are always on opposite sides of the circular boundary.

Surprisingly, a single vortex inside an elliptical boundary is considerably more complicated because the inverse Joukowsky transformation $z_\pm(w) = w \pm \sqrt{w^2-1}$ has a branch cut along the focal line joining $w = \pm 1$. This branch cut does not affect a vortex outside an elliptical boundary because the allowed region excludes the singularity. In contrast, the branch cut lies in the middle of the allowed region for a vortex inside an elliptical boundary, and a direct use of Eq. (28) to describe a vortex inside an ellipse would lead to a discontinuous potential along the focal line, as seen inside the translucent white region in Fig. 1.

As mentioned in Sec. 4, the Joukowsky map (27) has inversion symmetry for $z \leftrightarrow z^{-1}$, which has profound consequences for a vortex inside an elliptical boundary. Specifically, any conformal transformation involving a vortex inside a circular boundary $R$ must retain the same symmetry, so that the allowed region in the $z$ plane becomes the interior of an annulus bounded by concentric circles $R^{-1}$ and $R$. The left panel of Fig. 3 shows this annulus in the $z$ plane with black outer boundary at $R$ and black inner boundary at $R^{-1}$. The black dashed line has unit radius and splits the annulus into two separate regions; yellow for $|z| > 1$ and blue for $|z| < 1$. A vortex at position $z_0$ in the yellow region and its inverse at $z_0^{-1}$ in the blue region both map to a single vortex at $w_0$ inside the elliptical boundary, as shown in the right panel of Fig. 3.

The Joukowsky map takes both the inner and outer circles of radii $R^{-1}$, $R$ to the ellipse with semiaxes $a = \frac{1}{2}(R+R^{-1})$ and $b = \frac{1}{2}(R-R^{-1})$. In addition, the circle $|z| = 1$ maps onto the flat ellipse that is a closed loop infinitesimally close to the focal line of the ellipse $w \in [-1,1]$.

In this Section we find the complex potential in the $z$ plane that maps into a vortex inside an ellipse via the (inverse) Joukowsky transformation. We then combine this result with the general formulas derived in Sec. 2 to obtain the total energy and the dynamics of a single vortex and a vortex dipole inside an elliptical boundary. In Appendix A we provide a different but surprisingly equivalent derivation of these results, based on a direct map from the unit circle to the ellipse.

## 6.1 Complex potential and flow field

Consider a (positive) vortex at position $w_0$ inside an ellipse with major semiaxis $a$ and minor semiaxis $b = \sqrt{a^2-1}$. The dimensionless circulation (in units of $\hbar/M$) around the elliptical boundary is $\Gamma_{\text{ext}} = +2\pi$, while the one along the flat ellipse (surrounding the focal line) is $\Gamma_{\text{FE}} = 0$. The analogous complex potential on the $z$ plane for a vortex in an annulus at $z_0 = w_0 + \sqrt{w_0^2-1}$ must lead to the same circulations along the corresponding boundaries. Hence the complex potential on the $z$ plane must feature a circulation $\Gamma(R) = 2\pi$ on the outer boundary of the annulus and $\Gamma(1) = 0$ on the unit circle.

Physically, the hydrodynamic flow for a vortex inside an elliptical boundary should be smooth throughout the interior because it is basically a quadrupolar distortion of that for a circular boundary. Hence we must require that the flow be smooth and continuous across the focal line. The focal line of the ellipse on the $w$ plane is the image of the unit circle on the $z$ plane. A smooth potential on the ellipse therefore requires an extra boundary condition on the $z$ plane: on the unit circle, we require $F\left(z = e^{i\theta}\right) = F\left(z = e^{-i\theta}\right)$. In this way, when the Joukowsky transformation maps the unit circle onto the focal line, the potential will remain

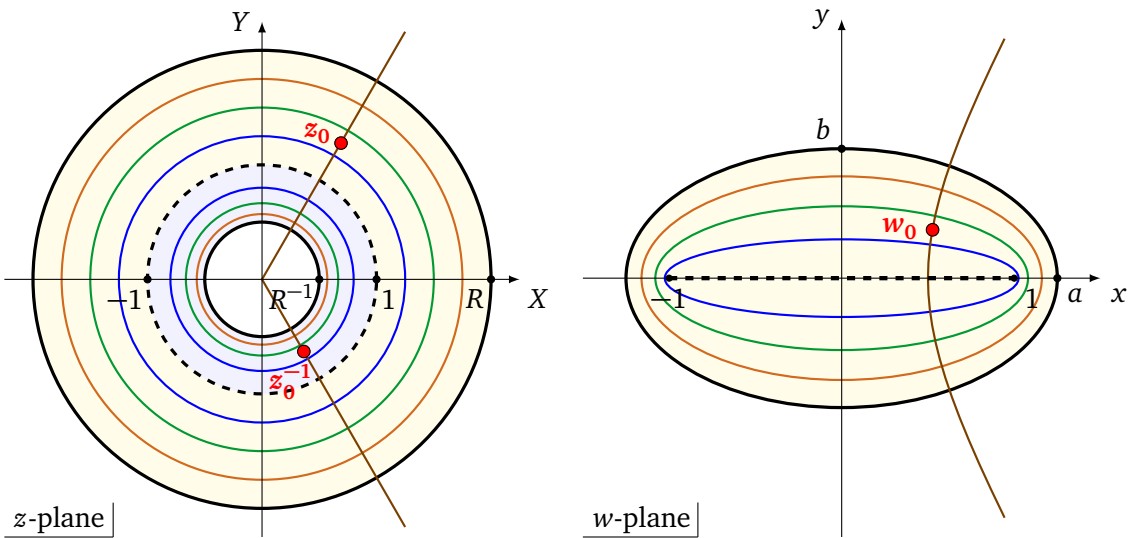

**Figure 3:** The Joukowsky transformation maps concentric circles into confocal ellipses. Curves with the same colour are related by the conformal transformation. The outer boundary at $R$ and the inner boundary at $R^{-1}$ form an annulus on the $z$ plane. The unit circle (black-dashed) is mapped into the degenerate flat ellipse surrounding the focal line. Yellow and blue-shaded regions in the left panel represent, respectively, the two Riemann sheets $|z| > 1$ and $|z| < 1$ within the annulus: each of these regions maps into the interior of the ellipse in the $w$ plane (right panel), so that the same ellipse maps back to two corresponding circles with inverse-symmetric radii. Similarly, the Joukowsky transformation sends the two points $z_0$ and $z_0^{-1}$ to a single point $w_0$. The ensemble of points with polar coordinates $\pm\theta_0$ inside the annulus, i.e. two semi-lines in the $z$ plane, is mapped to one branch of a hyperbola in the $w$ plane (brown curves).

smooth across the focal line. On the unit circle where $|z| = 1$, the latter expression is equivalent to $F(z) = F(z^{-1})$. We therefore seek a potential such that $F(z) = F(z^{-1})$ everywhere, which will automatically satisfy the extra requirement at $|z| = 1$.

Moreover, as seen in Fig. 3, on the $z$ plane we also need to impose two hard-wall boundary conditions: not only at $|z| = R$, but also at $|z| = R^{-1}$. In other words, we need to solve the problem on an annulus with inner and outer radii $R_{\text{in}} = R^{-1}$ and $R_{\text{out}} = R$.

The complex potential for a single (positive) vortex at position $z_0$ inside an annulus of radii $R^{-1} < 1 < R$, as derived in Refs. [11, 12], reads

$$F_{\text{annulus,single}}(z; z_0) = \ln\left[\frac{\vartheta_1\left(-\frac{i}{2}\ln\left(\frac{z}{z_0}\right), q\right)}{\vartheta_1\left(-\frac{i}{2}\ln\left(\frac{zz_0^*}{R^2}\right), q\right)}\right], \tag{32}$$

where $\vartheta_1(z, q)$ denotes the first Jacobi theta function, and its *nome* $q \equiv R_{\text{in}}/R_{\text{out}} = R^{-2} < 1$ can be rewritten in terms of the parameters of the ellipse as $q = (a-b)^2$. In particular, this potential yields a flow with circulation on the outer boundary only: i.e. $\Gamma(R) = 2\pi$, and $\Gamma(R^{-1}) = 0$.

Then, the simplest recipe to obtain a potential symmetric under the exchange $z \leftrightarrow z^{-1}$ is to write

$$F_{\text{annulus}}(z; z_0) = F_{\text{annulus,single}}(z; z_0) + F_{\text{annulus,single}}\left(z^{-1}; z_0\right). \tag{33}$$

To understand the physical meaning of this complex potential, we use the quasiperiodicity

of the Jacobi $\vartheta_1$ function[1] to show that (apart for an irrelevant constant) Eq. (33) may be conveniently rewritten as

$$F_{\text{annulus}}(z; z_0) = F_{\text{annulus,single}}(z; z_0) + F_{\text{annulus,single}}\left(z; z_0^{-1}\right) - \ln(z). \tag{34}$$

The first two terms in the latter equation correspond to the flows generated by a positive vortex at $z_0$ and by its positive symmetric partner located at $z_0^{-1}$. Each of these terms induces $2\pi$ circulation at the outer boundary $|z| = R$, and 0 at the inner one $|z| = R^{-1}$. The last term in the equation represents a negative vortex at the origin of the $z$ plane, which removes $2\pi$ circulation everywhere. As a result, the combination of the three terms yields $\Gamma(R) = 2\pi$ and $\Gamma(R^{-1}) = -2\pi$. Furthermore, the unit circle contains two vortices with opposite sign, so that the circulation along it vanishes, namely $\Gamma(1) = 0$. This behaviour reflects the built-in symmetry $z \leftrightarrow z^{-1}$, ensuring that $F_{\text{annulus}}\left(e^{i\theta}\right) = F_{\text{annulus}}\left(e^{-i\theta}\right)$. Summarizing, the complex potential (33) satisfies all the requested conditions [the circulations $\Gamma(R) = +2\pi$, $\Gamma(1) = 0$ and the symmetry which guarantees continuity across the flat ellipse] in the simply connected region $1 \le |z| \le R$. In this way, the Joukowsky map projects the latter region in a one-to-one fashion onto the whole ellipse, with no ambiguity.

The panels on the left side of Fig. 4 show the streamlines $\chi(\boldsymbol{r}) = \operatorname{Re} F_{\text{annulus}}(z; z_0)$, the phase field $\Phi(\boldsymbol{r}) = \operatorname{Im} F_{\text{annulus}}(z; z_0)$ and the superfluid velocity field $\boldsymbol{v}(\boldsymbol{r})$ [obtained from Eq. (2)] of this two-vortex configuration on the annulus in the $z$ plane.

As a final step, it is convenient to introduce the compact notation from Eq. (28)

$$\mathfrak{z} \equiv \mathfrak{z}(w) = w + \sqrt{w^2 - 1}, \qquad \mathfrak{z}_0 \equiv \mathfrak{z}(w_0). \tag{35}$$

As a result, the complex potential for a single vortex inside the ellipse in the $w$ plane has the following analytical expression:

$$
\begin{aligned}
F_{\text{ellipse}}(w; w_0) &= F_{\text{annulus}}(z = \mathfrak{z}; z_0 = \mathfrak{z}_0) \\
&= \ln\left[ \frac{\vartheta_1\left(-\frac{i}{2}\ln\left(\frac{\mathfrak{z}}{\mathfrak{z}_0}\right), q\right)}{\vartheta_1\left(-\frac{i}{2}\ln\left(q\mathfrak{z}\mathfrak{z}_0^*\right), q\right)} \right] + \ln\left[ \frac{\vartheta_1\left(-\frac{i}{2}\ln\left(\frac{1}{\mathfrak{z}\mathfrak{z}_0}\right), q\right)}{\vartheta_1\left(-\frac{i}{2}\ln\left(q\frac{\mathfrak{z}_0^*}{\mathfrak{z}}\right), q\right)} \right].
\end{aligned}
\tag{36}
$$

The result (36) satisfies the correct boundary conditions of the superfluid flow in the $w$ plane of the ellipse as it is clear in the panels on the right of Fig. 4, showing the streamlines, phase field and superfluid velocity field based on the complex potential (36).

We recall that a single vortex inside an annulus requires an infinite set of image vortices [11] to ensure that the superfluid flow is tangent at both the boundaries. The same situation holds for a single vortex inside an elliptical domain, as it is clear from our previous reasoning, based on the combination of the complex potential for an annulus with the Joukowsky transformation. In particular, we have verified that the infinite set of image vortices generated (outside the elliptical container) by Eq. (36) agrees with the set of images found in Ref. [28]. Our derivation, however, seems simpler and more straightforward, since it involves only the three Eqs. (32, 33, 36).

## 6.2 Total energy for a single vortex

The total energy for a vortex inside an ellipse may be computed very simply using the relation (9) which connects the energy on two surfaces linked by a conformal transformation. In our case, the $w$ plane has the elliptical boundary with semiaxes $a$ and $b$, and the $z$ plane has the annulus with inner and outer radii $R^{-1}$ and $R$.

---

[1] $\vartheta_1(z \pm \pi\tau, q) = -q^{-1}e^{\mp 2iz}\vartheta_1(z, q)$, where the parameter $\tau$ is related to the *nome* $q$ by $q \equiv e^{i\pi\tau}$.

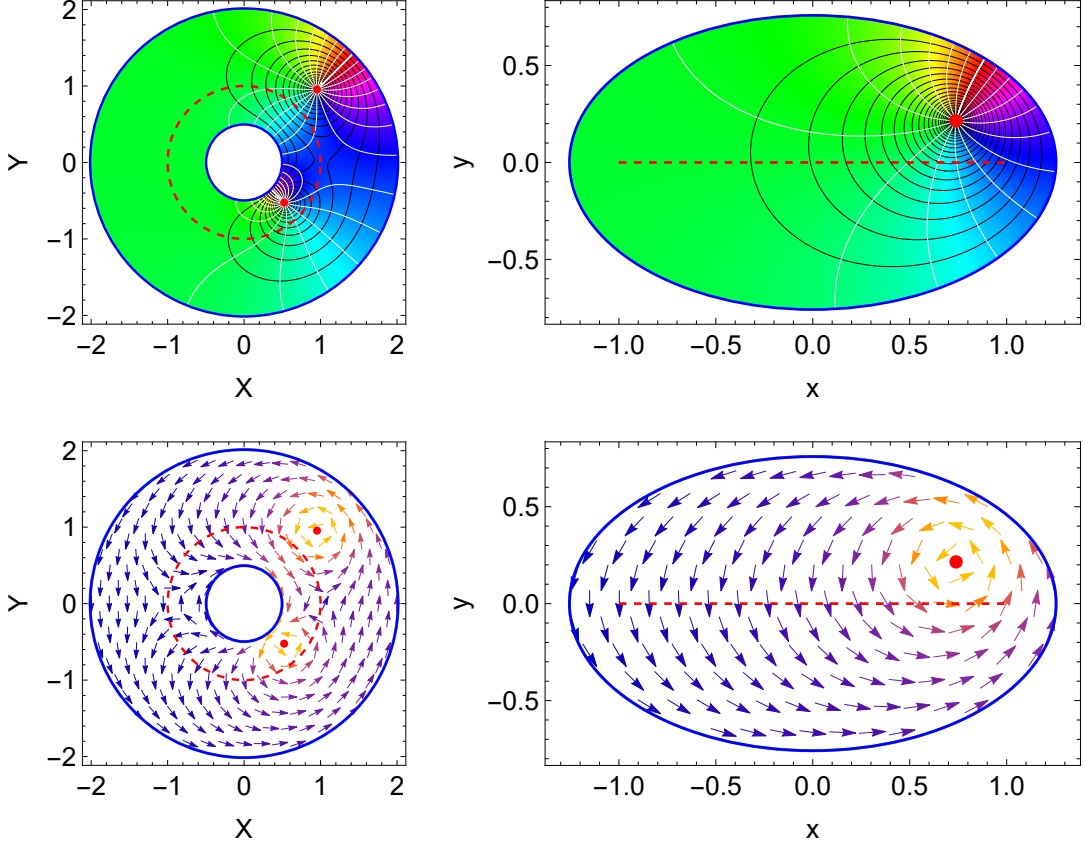

Figure 4: Left column: streamlines (black lines) and phase (colour coding and white lines) [top panel] and velocity field [bottom panel] associated with two vortices at positions $z_0$, $z_0^{-1}$ inside the planar annulus. The annulus has outer and inner radii $R = 2$, $R^{-1} = 1/2$, the dashed line representing the unit circle. The polar coordinates for the vortex at complex position $z_0 = r_0 e^{i\theta_0}$ are $r_0 = 1.35$, $\theta_0 = \pi/4$.
Right column: streamlines, phase and velocity field associated with a single vortex at $w_0$ inside the ellipse. The ellipse has eccentricity $b/a = 0.6$, and the cartesian coordinates of the vortex at position $w_0$ are $x_0 \approx 0.74$, $y_0 \approx 0.22$.

To compute the energy of a vortex in the annulus with the inverse map in Eq. (28), we must use only the "outer" region $1 \leq |z| \leq R$. This region contains a single vortex at position $z_0$ and the circulations around the inner ($r = 1$) and outer ($r = R$) boundaries are, respectively, $\Gamma_{\text{in}} = 0$ and $\Gamma_{\text{out}} = 2\pi\hbar/M$. The energy of the annulus is obtained from Eq. (8) as

$$E_0^{\text{annulus}} = \frac{\pi\hbar^2 n}{M}\left[(+1)\chi_{\text{out}} - 0\,\chi_{\text{in}} - \tilde{\chi}_0^z\right] = \frac{\pi\hbar^2 n}{M}\left(\chi_{\text{out}} - \tilde{\chi}_0^z\right). \tag{37}$$

We recall that the stream function is the real part of the complex potential (33). First, we compute the constant boundary term

$$\chi_{\text{out}} = \text{Re}\left[F_{\text{annulus}}\left(z = Re^{i\theta}; z_0\right)\right] = \ln(\sqrt{q}|z_0|). \tag{38}$$

After that, using

$$\lim_{z \to z_0} \ln\left[\vartheta_1\left(-\frac{i}{2}\ln\left(\frac{z}{z_0}\right), q\right)\right] = \lim_{z \to z_0} \ln(z - z_0) - \ln\left[-\frac{2z_0}{i\vartheta_1'(0, q)}\right],$$

the regularized stream function reads:

$$\tilde{\chi}_0^z = \mathrm{Re}\left\{\lim_{z\to z_0}\left[F_{\mathrm{annulus}}(z;z_0)-\ln(z-z_0)\right]\right\}$$

$$= -\ln\left[2|z_0|\frac{\vartheta_1\left(-\frac{i}{2}\ln\left(q|z_0|^2\right),q\right)}{i\vartheta_1'(0,q)}\right] - \mathrm{Re}\ln\left[\frac{\vartheta_1\left(-\frac{i}{2}\ln\left(q\frac{z_0^*}{z_0}\right),q\right)}{\vartheta_1\left(i\ln z_0,q\right)}\right]. \tag{39}$$

Substituting into Eq. (37), we obtain

$$\frac{E_0^{\mathrm{annulus}}}{\pi\hbar^2 n/M} = \ln\left[2\sqrt{q}|z_0|^2\frac{\vartheta_1\left(-\frac{i}{2}\ln\left(q|z_0|^2\right),q\right)}{i\vartheta_1'(0,q)}\right] + \mathrm{Re}\ln\left[\frac{\vartheta_1\left(-\frac{i}{2}\ln\left(q\frac{z_0^*}{z_0}\right),q\right)}{\vartheta_1\left(i\ln z_0,q\right)}\right]. \tag{40}$$

Finally, after recalling the scale factor $\sigma(w) = \ln\mathfrak{z}(w) - \ln\sqrt{w^2-1}$ and $\mathfrak{z}_0 = w_0 + \sqrt{w_0^2-1}$, Eq. (9) yields the total energy for a vortex inside an ellipse:

$$\frac{E_0^{\mathrm{ellipse}}}{\pi\hbar^2 n/M} = \frac{E_0^{\mathrm{annulus}}}{\pi\hbar^2 n/M} - \mathrm{Re}\,\sigma(w_0)$$

$$= \ln\left[2\sqrt{q}|\mathfrak{z}_0|\frac{\vartheta_1\left(-\frac{i}{2}\ln\left(q|\mathfrak{z}_0|^2\right),q\right)}{i\vartheta_1'(0,q)}\right] + \mathrm{Re}\ln\left[\sqrt{w_0^2-1}\frac{\vartheta_1\left(-\frac{i}{2}\ln\left(q\frac{\mathfrak{z}_0^*}{\mathfrak{z}_0}\right),q\right)}{\vartheta_1\left(i\ln\mathfrak{z}_0,q\right)}\right]. \tag{41}$$

For numerical calculations one needs to define the branch cut for the square root appearing in those expressions using:

$$\sqrt{w^2-1} \equiv \sqrt{|w-1||w+1|}\exp\left[\frac{i}{2}(\arg(w-1)+\arg(w+1))\right].$$

Some constant-energy contours are shown in the left panel of Fig. 5.

## 6.3  Vortex trajectories

The velocity of a vortex at position $z_0$ given by the complex potential (33) is:

$$i\dot{z}_0^* = \frac{\hbar}{M}\lim_{z\to z_0}\left[\frac{dF_{\mathrm{annulus}}(z;z_0)}{dz} - \frac{1}{z-z_0}\right]$$

$$= \frac{\hbar}{M}\frac{i}{2z_0}\left[i + \frac{\vartheta_1'\left(i\ln z_0,q\right)}{\vartheta_1\left(i\ln z_0,q\right)} + \frac{\vartheta_1'\left(-\frac{i}{2}\ln\left(q|z_0|^2\right),q\right)}{\vartheta_1\left(-\frac{i}{2}\ln\left(q|z_0|^2\right),q\right)} - \frac{\vartheta_1'\left(-\frac{i}{2}\ln\left(q\frac{z_0^*}{z_0}\right),q\right)}{\vartheta_1\left(-\frac{i}{2}\ln\left(q\frac{z_0^*}{z_0}\right),q\right)}\right]. \tag{42}$$

Then, the velocity for a vortex at position $w_0$ inside the ellipse follows directly from Eq. (13) as:

$$i\dot{w}_0^* = \frac{\hbar}{2M}\left\{-\frac{w_0}{w_0^2-1} + \frac{i}{\sqrt{w_0^2-1}}\left[\frac{\vartheta_1'\left(i\ln\mathfrak{z}_0,q\right)}{\vartheta_1\left(i\ln\mathfrak{z}_0,q\right)} + \right.\right.$$

$$\left.\left. + \frac{\vartheta_1'\left(-\frac{i}{2}\ln\left(q\,|\mathfrak{z}_0|^2\right),q\right)}{\vartheta_1\left(-\frac{i}{2}\ln\left(q\,|\mathfrak{z}_0|^2\right),q\right)} - \frac{\vartheta_1'\left(-\frac{i}{2}\ln\left(q\frac{\mathfrak{z}_0^*}{\mathfrak{z}_0}\right),q\right)}{\vartheta_1\left(-\frac{i}{2}\ln\left(q\frac{\mathfrak{z}_0^*}{\mathfrak{z}_0}\right),q\right)}\right]\right\}. \tag{43}$$

We numerically integrated these complex dynamical equations for various initial conditions along the positive real axis, giving the closed trajectories shown in the right panel of Fig. 5. As proved in Sec. 2.2.1, these orbits are also contours of constant energy, and we checked that they

indeed coincide with the curves shown in the left panel of Fig. 5. Note that these orbits differ greatly from the confocal ellipses in the right panel of Fig. 3. Instead they resemble nested self-similar ellipses, which is not surprising because the elliptical boundary is a quadrupolar distortion of a circle. Thus the closed trajectories for the elliptical boundary should resemble a set of nested circles with quadrupolar distortions.

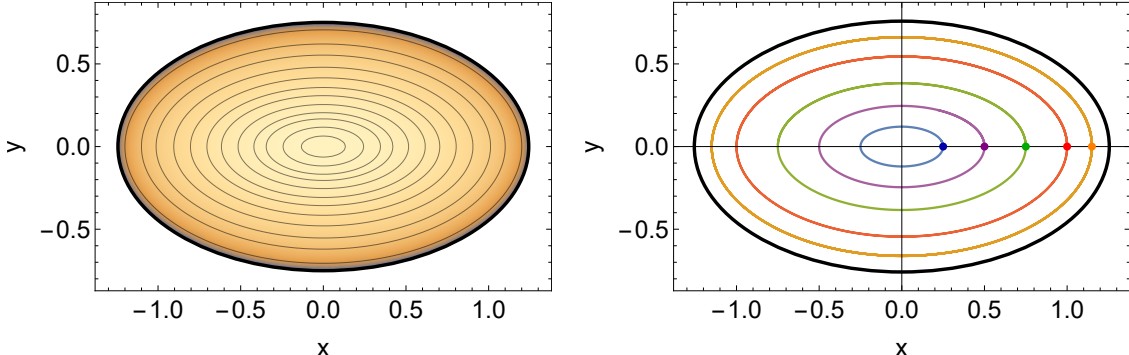

Figure 5: Left: constant-energy contours for a vortex inside an ellipse with aspect ratio $b/a = 0.6$. The thick black line denotes the outer boundary with $a = 5/4$ and $b = 3/4$. The vortex energy is negative and it decreases moving from the centre towards the boundary. Right: trajectories of a single vortex inside the same elliptical boundary for various initial positions on the positive horizontal axis. Coloured dots denote the starting point of the different trajectories, which are obtained from numerical solution of the equations of motion (43).

In detail, however, the situation is more complicated. For a circular boundary, the rotational symmetry ensures conservation of angular momentum, so that all vortex trajectories are circles. In contrast, an elliptical boundary does not conserve angular momentum because the boundary is invariant only under a finite rotation by $\pm\pi$. If we use an angular Fourier expansion for the trajectory, only even harmonics can occur, and the nonlinear dynamical equations couple the various Fourier amplitudes. As a result, the orbits are not strictly ellipses. More quantitatively, for a given trajectory we extract the maximum value of the coordinates $(x_{max}, y_{max})$ and the period $T$. Figure 6 shows $(x_0/x_{max})^2 + (y_0/y_{max})^2$ evaluated along the five closed trajectories in the right frame of Fig. 5. The quantity $(x_0/x_{max})^2 + (y_0/y_{max})^2$ deviates from the value 1 that it would have for a pure ellipse, displaying a periodic, anharmonic dependence on the polar angle $\theta_0 \in [0, 2\pi]$. As follows from its definition, the maximum value of 1 is reached at positions $(\pm x_{max}, 0)$ and $(0, \pm y_{max})$. The minimum value and the corresponding deviation from 1 differ for each case, but it always remains small (less than 1%). Starting from the smallest trajectory (blue curve), the amplitude of the oscillation increases (purple and green), until it reaches its maximum for the trajectory with $x_{max} = 1$ (red), which is the one passing through the *foci* of the ellipse. Then, for $1 < x_{max} < a$ (orange curve), deviations from 1 progressively decrease due to the influence of the elliptic boundary.

The numerical evaluation of the period $T$ for each trajectory then gives the mean precession frequency $\langle\Omega\rangle = 2\pi/T$. Figure 7 shows $\langle\Omega\rangle$ as a function of the initial position along the positive horizontal axis, for three values of the aspect ratio of the boundary. As the aspect ratio approaches 1, the mean precession frequency converges to the red solid curve, which corresponds to the result for a single vortex inside a disk of radius $a$ [see Eq. (22)], namely $\Omega(x_0) = (\hbar/M)(a^2 - x_0^2)^{-1}$. Importantly, the mean angular velocity is always positive, hence a positive vortex can only move in the counterclockwise direction, independent of its initial position. This result is similar to the circular boundary, while it differs from the case of a vortex inside an annulus, where the precession frequency can be both positive and negative,

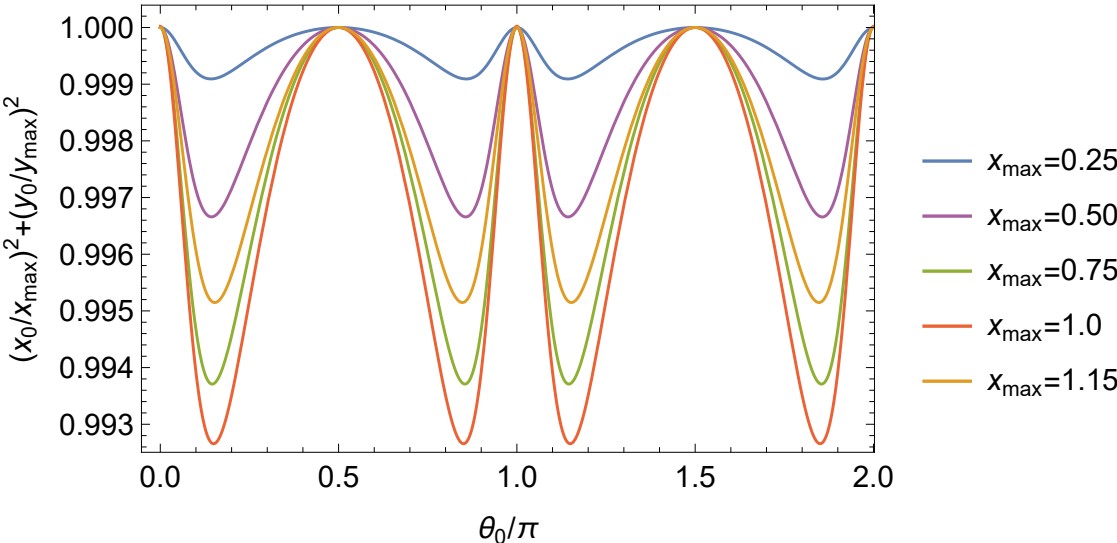

Figure 6: Quantity $(x_0/x_{max})^2 + (y_0/y_{max})^2$ as a function of the polar angle $\theta_0$ along a closed vortex trajectory inside an ellipse with $b/a = 0.6$. The different curves refer to five trajectories with various initial positions $(x_0(0), y_0(0)) = (x_{max}, 0)$ on the positive horizontal axis. Each case shows a periodic deviation from the value 1 that is expected for a pure ellipse.

depending on its position relative to the inner and outer boundaries. Physically, this latter behaviour reflects the influence of the nearest image, which lies beyond the closest boundary.

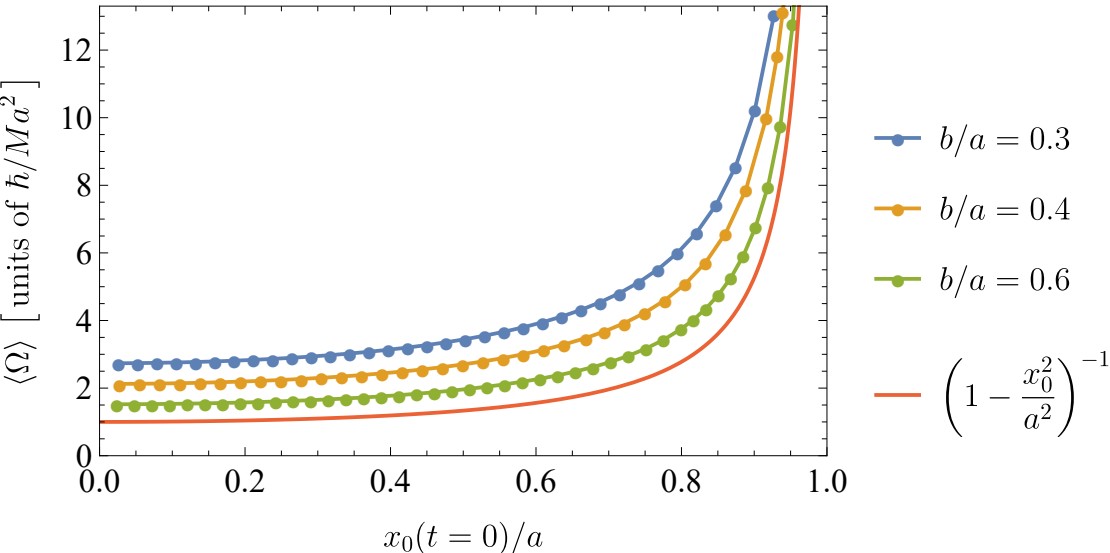

Figure 7: Dependence of the mean precession frequency on the initial position along the positive horizontal axis for three different aspect ratios $b/a$ of the elliptical boundary. The red solid curve is the result for a circular boundary of radius $a$.

## 6.4 Example of a multivortex configuration: a symmetric dipole

We now summarize the straightforward generalization to $N_v$ vortices at positions $w_j = w(z_j)$ and with charges $c_j$ ($j = 1, \ldots, N_v$). We recall that, for a single positive vortex inside the ellipse,

the complex potential in the $z$ plane is given by Eq. (33). Now, we can apply the superposition principle and start with the following complex potential:

$$F_{\text{annulus}}(z; \{z_j\}) = \sum_{j=1}^{N_v} c_j \left[ F_{\text{annulus,single}}(z; z_j) + F_{\text{annulus,single}}(z^{-1}; z_j) \right].$$ (44)

To compute the total energy, we evaluate first the constant value of the stream function along the outer boundary of the annulus

$$\chi_{\text{out}} = \text{Re} \, F_{\text{annulus}} \left( z = R e^{i\theta}; \{z_j\} \right),$$ (45)

and then the regularized stream function at the $j^{th}$ vortex core

$$\tilde{\chi}_j^z = \text{Re} \lim_{z \to z_j} \left[ F_{\text{annulus}}(z; \{z_j\}) - c_j \ln(z - z_j) \right].$$ (46)

The $z \leftrightarrow z^{-1}$ symmetry ensures that the circulation around the boundary $|z| = 1$ is equal to zero, while the circulation around the outer boundary is $\Gamma_{\text{out}} = \sum_{j=1}^{N_v} 2\pi\hbar c_j/M$. A generalization of Eq. (37) gives the total energy of these $N_v$-vortices in the $1 \leq |z| \leq R$ region:

$$\frac{E_{N_v}^{\text{annulus}}}{\pi\hbar^2 n/M} = \sum_{j=1}^{N_v} c_j \left( \chi_{\text{out}} - \tilde{\chi}_j^z \right).$$ (47)

The total energy in the $w$ plane is finally obtained from Eq. (17) as:

$$\frac{E_{N_v}^{\text{ellipse}}}{\pi\hbar^2 n/M} = \frac{E_{N_v}^{\text{annulus}}}{\pi\hbar^2 n/M} - \sum_{j=1}^{N_v} c_j^2 \, \text{Re} \, \sigma(w_j).$$ (48)

We now consider a vortex dipole oriented symmetrically with respect to the focal line:

$$w_1 = w_0, \qquad w_2 = w_0^*, \qquad c_1 = -c_2 = 1.$$

Equation (44) gives the complex potential from which one can easily compute the streamlines and phase field shown in the left panel of Fig. 8.

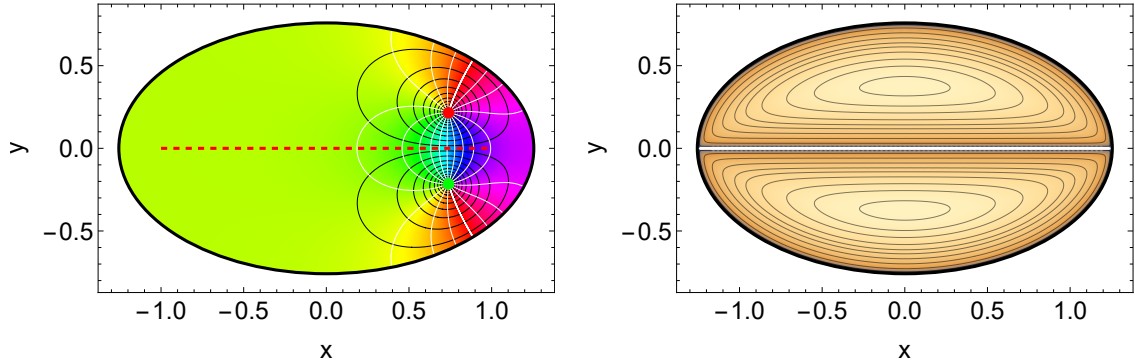

Figure 8: Left: streamlines (black lines) and phase (colour coding and white lines) associated with a symmetric vortex dipole inside the ellipse. This configuration consists of a positive vortex at position $w_0$ (red dot) and a negative one that is symmetric with respect to the focal line at position $w_0^*$ (green dot). Right: constant-energy contours for the same symmetric vortex dipole. Colour coding is the same as in Fig. 5. The elliptical boundary has aspect ratio $b/a = 0.6$.

The total energy of this configuration (as a function of cartesian coordinates $\{x_0, y_0\}$) comes out from Eq. (48) as:

$$
\begin{aligned}
\frac{E_{\text{dipole}}^{\text{ellipse}}}{\pi \hbar^2 n/M} =&\, 2\ln\left[\frac{2}{i}\frac{\vartheta_1\left(-\frac{i}{2}\ln\left(q|\mathfrak{z}_0|^2\right),q\right)}{\vartheta_1\left(\frac{i}{2}\ln q,q\right)}\frac{\vartheta_1\left(-\frac{i}{2}\ln\left(|\mathfrak{z}_0|^2\right),q\right)}{\vartheta_1'(0,q)}\right] \\
&+ 2\,\mathrm{Re}\ln\left[\sqrt{w_0^2-1}\,\frac{\vartheta_1\left(-\frac{i}{2}\ln\left(q\frac{\mathfrak{z}_0^*}{\mathfrak{z}_0}\right),q\right)}{\vartheta_1\left(\frac{i}{2}\ln\left(q\mathfrak{z}_0^2\right),q\right)}\frac{\vartheta_1\left(-\frac{i}{2}\ln\left(\frac{\mathfrak{z}_0^*}{\mathfrak{z}_0}\right),q\right)}{\vartheta_1\left(\frac{i}{2}\ln\left(\mathfrak{z}_0^2\right),q\right)}\right].
\end{aligned}
\tag{49}
$$

A plot of constant-energy contours is shown in the right panel of Fig. 8. The trajectories are qualitatively similar to the ones described by a vortex dipole inside a circular boundary, which have been experimentally observed in Ref. [32].

## 7  Conclusions and outlook

An earlier paper by one of us [18] studied the low-lying equilibrium states of rotating superfluid He-II in an elliptical cylinder, using real elliptic coordinates with infinite series for the resulting analytic solutions. Here, instead, we relied on complex variables and conformal maps that permit a unified description of general bounded and curved two-dimensional surfaces. Reference [30] developed this formalism to relate the energy of vortices on a complicated surface to the corresponding energy on a simple surface through the metric of the conformal map connecting them. Here we demonstrated that a similar derivation yields the dynamical equations of vortices on general two-dimensional surfaces with boundaries. We also verified that Hamilton's equations (based on the energy of the vortex) reproduce the vortex dynamics obtained with our complex formalism (based on conformal maps).

A circular boundary with radius $R$ served as our model system because it is easy to derive the dynamics and energy of a vortex on either side of that boundary. To study vortex dynamics for an elliptical boundary, we used the Joukowsky map that takes a family of concentric circles into a family of confocal ellipses. This map worked well for a single vortex outside an elliptical boundary, giving the complex potential and the vortex orbits that are also the constant-energy curves.

The problem of a vortex inside an elliptical boundary became more intricate because the Joukowsky map introduced a branch cut along the line joining the two focal points of the ellipse. When combined with the inversion symmetry of the Joukowsky transformation, as seen in Fig. 3, the single circular boundary became an annulus with inner and outer boundaries at $R^{-1}$ and $R$. The application of the Joukowsky map then provided the complex potential for a vortex inside an elliptical boundary, along with physical properties like the energy and trajectories. A straightforward generalization for multiple vortices allowed us to study the behaviour of a symmetric vortex dipole inside an elliptical boundary.

After this work was nearly completed, we belatedly discovered an earlier different conformal map from the interior of a circle to the interior of an ellipse. This transformation, due to Schwarz (1869), involved Jacobian elliptic functions. We discuss its properties in Appendix A, where we verify that this map yields the same results found with the Joukowsky map.

As a future extension of this work, one could investigate how a finite massive core affects the vortex dynamics. The time-dependent variational Lagrangian method successfully described massive vortices in a disk [33–35] and in an annulus [36]. It would be interesting to verify that a massive vortex in an elliptical boundary exhibits similar behaviour. Another natural extension of our research would be to study vortex lattices in elliptical containers, and their normal modes and instabilities.

## Acknowledgements

**Funding information**  A. R. received funding from the European Union's Horizon research and innovation programme under the Marie Skłodowska-Curie grant agreement *Vortexons* (no. 101062887). P. M. and A. R. acknowledge support by the Spanish Ministerio de Ciencia e Innovación (MCIN/AEI/10.13039/501100011033, grant PID2020-113565GB-C21), and by the Generalitat de Catalunya (grant 2021 SGR 01411). P. M. further acknowledges support by the *ICREA Academia* program. M. C., A. R. and P. M. would like to thank the Institut Henri Poincaré (UAR 839 CNRS-Sorbonne Université) and the LabEx CARMIN (ANR-10-LABX-59-01) for their support.

# A  Flow inside an ellipse via direct mapping from the circle

We provide here an alternative and equivalent derivation of the complex potential, energy and velocity of a vortex inside an ellipse, based on the direct conformal transformation from a circle to an ellipse.

## A.1  Derivation of the map circle $\longleftrightarrow$ ellipse

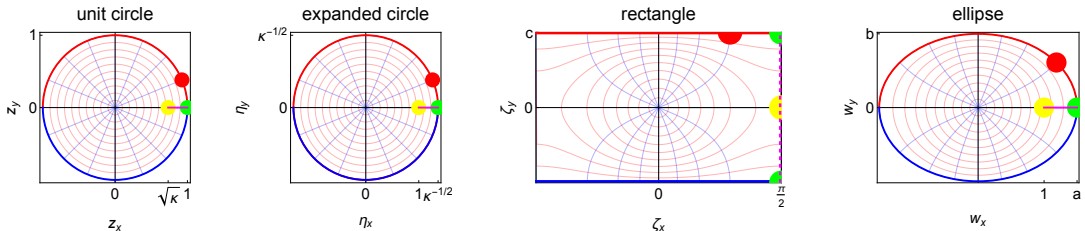

Figure 9:   The three conformal transformations mapping the unit circle on the $z$ plane (panel 1) to a bigger circle on the $\eta$ plane (panel 2) to a rectangle on the $\zeta$ plane (panel 3) and finally to an ellipse on the $w$ plane (panel 4). Dots of identical colours and the red, blue and magenta segments are sent to each other by the various maps. The map $\zeta(\eta)$ has a branch cut along the magenta segment (from $\eta = 1$ to $\eta = 1/\sqrt{\kappa}$), but this branch cut is glued seamlessly when one inserts $\zeta(\eta)$ into $w(\zeta)$.

The map from the unit circle on the $z$ plane to an ellipse with major and minor semi-axes $a$ and $b$ (and foci at positions $w = \pm 1$) on the $w$ plane was found by Schwarz in 1869 [37], and much later also re-derived in Ref. [38]. Here we outline a simple derivation of it. Consider the following three conformal transformations:

1. $\eta(z) = z/\sqrt{\kappa}$ stretches the plane radially. We will see that $0 < \kappa < 1$, so the unit circle becomes a bigger circle of radius $1/\sqrt{\kappa}$.

2. $\zeta(\eta) = \alpha F\left(\arcsin(\eta)|\kappa^2\right)$ takes a circle with radius $1/\sqrt{\kappa}$ to a rectangle centered at the origin, with its top-right corner at complex position $\alpha(K + iK')$. Here $\alpha$ is a number, $F(\phi|m)$ is the elliptic integral of the first kind with *parameter $m$* (and *modulus $\kappa = \sqrt{m}$*), $K \equiv F\left(\frac{\pi}{2}|m\right)$ is the corresponding complete integral, and $K' \equiv F\left(\frac{\pi}{2}|m'\right)$ with $m' = 1-m$ the complementary parameter. We set $\alpha = \frac{\pi}{2K}$, so that the top-right corner of the rectangle appears at $\zeta = \frac{\pi}{2} + ic$, with $c = \frac{\pi}{2}\frac{K'}{K}$. The aspect ratio $\frac{K'}{K}$ of this rectangle yields the *nome $q$* through $q = \exp\left[-\pi\frac{K'}{K}\right] = e^{-2c}$. This map is a variant of the well-known Schwarz-Christoffel map [39].

3. $w(\zeta) = \sin(\zeta)$ maps a rectangle with top-right corner at $\zeta = \pi/2 + ic$ to an ellipse with semi-axes $a = \cosh(c)$, $b = \sinh(c)$, and foci at $w = \pm 1$. Noting that $\sin(\zeta) = i\sinh(-i\zeta)$, this map is a rotated version of the familiar relation for the elliptic coordinates $w = \sinh(\zeta)$.

The action of the three maps is shown in Fig. 9. The transformation from the circle to the ellipse is now obtained very simply concatenating the three maps:

$$w(z) = w(\zeta(\eta(z))) = \sin\left(\frac{\pi F\left(\arcsin\left(\frac{z}{\sqrt{\kappa}}\right)\middle|\kappa^2\right)}{2K}\right), \tag{A.1}$$

where we recall that $K \equiv F\left(\frac{\pi}{2}\middle|\kappa^2\right)$. The *modulus* $\kappa$ of the transformation controls the aspect ratio of the final ellipse. It has to be found imposing that the transformation sends $z = 1$ to $w = a$ [see the green dot in Fig. 9], which amounts to solving

$$w(\zeta(\eta(z)))|_{z=1} = a. \tag{A.2}$$

Quite surprisingly, the latter equation has an analytical solution. Consider the known relation

$$\kappa \equiv \left(\frac{\vartheta_2(0, q^2)}{\vartheta_3(0, q^2)}\right)^2, \tag{A.3}$$

which gives the modulus $\kappa$ in terms of the nome $q$ of Jacobi Theta functions. The last step is to realize that for an ellipse with foci at $w = \pm 1$ we have

$$q = e^{-2c} = [\cosh(c) - \sinh(c)]^2 = (a - b)^2 = (a - b)/(a + b), \tag{A.4}$$

which implies that $\kappa$ is given by Eq. (A.3) with $q \equiv (a - b)/(a + b)$. Since $0 < q < 1$, it follows that $0 < \kappa < 1$.

## A.2 Complex potential, energy and core velocity of a single vortex

With the conformal map in Eq. (A.1), it is now straightforward to derive the properties of a single vortex inside an ellipse. Using $F(\arcsin(v)|m) = \text{arcsn}(v|m)$ and $K = \text{arcsn}(1|m)$, where $\text{arcsn}(v|m)$ denotes the inverse of Jacobi's elliptic function $\text{sn}(v|m)$, the inverse map from the ellipse to the unit circle is found to be

$$z(w) = \sqrt{\kappa}\,\text{sn}\left(\frac{2K\,\arcsin(w)}{\pi}\middle|\kappa^2\right). \tag{A.5}$$

The complex potential for a vortex at $w_0$ inside an ellipse is then obtained from Eq. (23) as

$$F_{\text{ellipse,dir}}(w; w_0) = \ln\left[\frac{z(w) - z(w_0)}{z(w) - 1/z(w_0)^*}\right], \tag{A.6}$$

with $z(w)$ given by Eq. (A.5).

To obtain the energy of a vortex inside the ellipse, we first compute the energy of a vortex in the unit circle. Setting $R = 1$ and $\mathfrak{n}_0 = 1$ in Eq. (26) we obtain $E_{\text{circle}} = \frac{\pi\hbar^2 n}{M}\ln\left(1 - |z_0|^2\right)$. Using Eq. (9), the energy for a vortex at $w_0$ inside the ellipse then follows immediately,

$$E_{\text{ellipse}} = E_{\text{circle}} - \frac{\pi\hbar^2 n}{M}\text{Re}\,\sigma(w_0). \tag{A.7}$$

where the scale factor $\sigma(w)$ is computed inserting Eq. (A.5) into Eq. (4).

Similarly, the complex velocity of a vortex inside the unit circle is given by Eq. (21) and reads $i\dot{z}_0^* = \frac{\hbar}{M z_0}\left(\frac{|z_0|^2}{1-|z_0|^2}\right)$. The complex velocity of a vortex inside the ellipse readily follows using Eq. (13),

$$i\dot{w}_0^* = i\dot{z}_0^* \, e^{\sigma(w_0)} + \frac{\hbar}{2M}\sigma'(w_0). \tag{A.8}$$

Numerical evaluation of Eqs. (A.6), (A.7) and (A.8) shows that these expressions coincide precisely with Eqs. (36), (41) and (43), even though they look radically different.

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
