# Peer review of "Conformal maps and superfluid vortex dynamics on curved and bounded surfaces: the case of an elliptical boundary"

_SciPost Physics, doi:SciPost Phys. 17, 039 (2024)_

## Round 1 · Referee Report · Anonymous (Referee 1) · 2024-2-22

Strengths

  1. Novel approach to solving for the energy and related quantities for a vortex inside an ellipse.
  2. Interesting numerical results demonstrating the slightly non-elliptical trajectories.
  3. Well written and logically structured.

Weaknesses

  1. Minimal new physics.
  2. Some mathematical arguments could be clarified.

Report

This work studies the dynamics and energetics of a vortex in a two-dimensional superfluid confined within an elliptical boundary. The authors combine recent results for the energy of a vortex in an annular potential with a Joukowsky transformation to derive closed form expressions for the energy and equations of motion of a vortex in an elliptical container. Although an analytic expression for the energy has been derived previously, the newly presented derivation is argued to be simpler and more intuitive. The vortex trajectories resulting from the obtained expressions are presented, and it is found that they are slightly non-elliptical. The mean vortex orbital frequency is numerically measured, and is found to approach the known solution for a disk as the ellipse's aspect ratio approaches unity, as expected.

While this work presents a novel derivation of the behavior of a vortex inside an ellipse, I am not convinced that the manuscript satisfies the high acceptance criteria of SciPost Physics. This new derivation of a largely known result is not a groundbreaking discovery (criteria 1), nor a breakthrough on a long-standing research problem (criteria 2). It is not clear that it opens up new pathways for research (criteria 3), and I do not see how it links different research areas in a novel way (criteria 4). As such, I think this work is better suited for publication in a more specialised journal such as SciPost Physics Core.

Requested changes

I had a number of minor comments and questions for the authors to consider that may further improve the clarity and presentation of the manuscript:

  • The authors refer throughout the manuscript to their newly derived expressions as "closed form", in contrast to the previous result of Ref. [17], which involved an infinite sum. However, the new expressions involve the Jacobi theta function, which itself is defined as an infinite summation. So:
  • Is the new expression really "closed form", given that it involves infinite sums?
  • Regardless of the semantics, how is it a significant simplification compared to the previous result?

  • In paragraph 1 of the introduction, atomic gases are described as having "negligible compressibility". While the flow field may be incompressible if the atomic density is uniform, I would not describe the fluid itself as incompressible. It is a compressible fluid that can support both incompressible and compressible flows (ie. sound waves). Similarly, in paragraph 5, the authors state that a cold atom system in the Thomas-Fermi regime will support incompressible flow. However, again this is only true in a uniform system. As a counter-example, a BEC in a harmonic trap will have varying spatial density $n$ in the Thomas-Fermi regime, and hence $\nabla \cdot (n \mathbf{v}) \neq n (\nabla \cdot \mathbf{v})$, meaning that the flow is not incompressible. I suggest the authors change the wording in these two places.

  • In the left frame of Fig. 2, I suggest labelling the angles $+\theta_0$ and $-\theta_0$ to emphasise that the angular positions of the two vortices are reflections about the X axis. It is also a bit confusing that the example configuration in the z-plane has the two vortices separated by ~90 degrees, since this is not the case in general. Using a different angle as the example might improve the clarity of the figure.

  • In the second sentence of Sec. 3, the authors quote from Ref. [33]: "in flow patterns related by a conformal map, circulation integrals around corresponding curves are the same". Looking at Fig. 2, I am confused by this. If one takes an integral around the orange path in the w-plane, the winding is $+2\pi$ since there is a single vortex enclosed. Likewise, taking an integral around the outer orange path in the z-plane gives $+2\pi$ (since this path encloses two positive vortices, plus a negative vortex at the origin of the z-plane, as described around Eq. 9). However, integrating around the inner orange path in the z-plane only encloses the central negative vortex, and hence the winding is $-2\pi$. Why do these windings around supposedly equivalent trajectories not agree?

  • In Sec. 3.1, the authors state that: "to ensure that the flow on the ellipse remains continuous across the branch cut, we must require $F_{annulus}(e^{i\theta}) = F_{annulus}(e^{-i\theta})$ on the unit circle". It is not clear to me why this makes the flow continuous along the branch cut. Could the authors expand on this?

  • Does the argument presented in Eqs. (8) and (9) extend to the case of a multiquantum vortex in the ellipse? Presumably then one would need a negative multiquantum vortex in the center of the ellipse to ensure there is no winding around the unit circle?

  • Eq. (12) is the equation of motion for a vortex in the ellipse. Can the Joukowsky transform be used to map this velocity to the velocity of the equivalent vortex in the annulus?

  • Immediately after Eq. (12), the authors state that they "integrate the complex dynamical equation (12)". It would be useful to the reader to specify here that the integration is done numerically (as stated in the introduction).

  • Regarding the results in Fig. 5, is there an intuitive reason why the trajectory is closest to being elliptical for the smallest orbital radius (blue curve)? Does the orbit become perfectly elliptical as the vortex approaches either the origin or the outer boundary of the ellipse?

  • In the final paragraph of Sec. 3.2, I suggest adding a citation for the quoted precession frequency of a vortex inside a disk.

  • Following Eq. (20), the authors assert that the vortex trajectories in Fig. 4 correspond to curves of constant energy. Could the authors plot these constant energy curves calculated from (20) in the figure, for comparison with the numerically integrated trajectories?

---

## Round 1 · Referee Report · Anonymous (Referee 2) · 2024-3-7

Strengths

  1. Well-written, clear and concise manuscript
  2. Of interest to a wide audience working in fluid flow in both classical and quantum fluids.
  3. Uses elements of conformal transformations to derive analytical results for more complex setups

Weaknesses

  1. Lack of significance of results
  2. Reformulates pre-existing results via a new method
  3. Not clear on the generality of the final results

Report

The manuscript investigates the dynamics of a point vortex inside a bounded elliptical domain. The key result of this work is that the authors use Joukowsky’s conformal transformation to map an annulus domain into an elliptical one whereby a simpler problem can be solved. This yields an alternative representation of the point vortex dynamics in an elliptical domain that the authors allude to being more concise than previous methods using infinite images. Moreover, the authors show that the resulting vortex evolution is of a form of a near elliptical orbit that exhibits a small periodic deviation, which are then investigated across the parameter ranges of the elliptical domain.

The manuscript is extremely well-written and is both clear and concise (general criteria 1). The work is relevant to a wide audience, especially those associated with vortex dynamics in fluid and turbulent flow of both classical and quantum fluids, where the use of point vortices as an approximation is frequently used. The results are interesting and provide new insight to those studying vortex dynamics in elliptical domains, such as those, alluded to by the authors, working in condensed matter BECs, where quasi-2D quantum turbulence is now a predominant focus in the community with the application of vertically confined traps. However, I struggle to see the significance of this work. It is not totally clear to me in what way their analytical results are simpler to previous results for the same problem. Moreover, as the study is for only one vortex only, do their results extend to any multi-vortex configuration? This means that I find it hard to attribute one of the expectation criteria for publication is SciPost Physics.

I am not confident that publication in SciPost Physics is warranted due to the lack if significance, breakthrough, or novelty of their results. Alternatively, a more niche.focussed journal may be more appropriate.

Requested changes

  1. Can the authors explain why their results are simpler than previous works in this setup? Is it the compact form of the analytical expressions or there is some computational element?

  2. Point vortex simulations can be notoriously finicky with regards to numerical convergence (particularly when simulating vast multi-vortex configurations). I think it would be prudent if the authors included some technical details on the specific numerical methods they use.

  3. Subsequently, the authors compute a formula for the total energy of the point vortex system in the elliptical domain and assure the reader that it is conserved. It seems prudent to ask the authors to compute this energy during their simulations to ensure that it is indeed conserved and to what degree?

  4. How does your results extend to multi-vortex configurations? Particularly to the argument of using equations (8) and (9).

  5. In Fig. 5, the amplitude of the periodic deviation decreases with the initial position of the vortex to the centre of the ellipse. Is there an explanation for this? How is it related to the position of x_0?

  6. I suggest that Fig. 5 should include a legend to help explain the different colour lines rather than referring to the legend of Fig. 4.

  7. The sentence just before equation (13) needs rewriting as I believe that the grammar is not correct.

  • validity: high
  • significance: ok
  • originality: high
  • clarity: top
  • formatting: excellent
  • grammar: excellent

Author:  Matteo Caldara  on 2024-06-24  [id 4581]

(in reply to Report 2 on 2024-03-07)
Category:
answer to question
reply to objection

We thank the Referee for their careful and constructive report. We also thank them for providing further useful comments and suggestions for improving our work. We have substantially modified our manuscript in order to highlight the novel results of our work and to fulfill the SciPost Physics acceptance criteria. We provide a detailed list of changes, together with a complete point-by-point reply, in the attached pdf document.

Attachment:

Reply_to_Referees.pdf

---

## Round 2 · Referee Report · Anonymous (Referee 3) · 2024-6-25

Strengths

1. Well-written, clear and concise manuscript
2. Of interest to a wide audience working in fluid flow in classical and quantum fluids.
3. Novel use of conformal transformations to derive analytical results for modelling complex vortex setups in bounded domains.

Report

The authors have submitted a revised manuscript that includes changes and responses to all of the reviewers’ comments. In doing so, they have adequately addressed the main criticism from both referees regarding the significance and novelty of the work and made significant improvements to the manuscript through revisions and additional research.

The revised introduction clarifies how the present work compares to previously published research. The revised manuscript significantly strides in using conformal transformations to study point vortex dynamics. It shows how the work can be applied to more general cases of point vortex dynamics in bounded domains.

Consequently, I am happy to recommend the publication of this work to SciPost Physics.

Requested changes

The Authors have addressed all changes previously outlined. No more changes are requested.

Recommendation

Publish (meets expectations and criteria for this Journal)

  • validity: top
  • significance: high
  • originality: high
  • clarity: top
  • formatting: perfect
  • grammar: excellent

Author:  Matteo Caldara  on 2024-07-18  [id 4629]

(in reply to Report 1 on 2024-06-25)
Category:
remark

We are grateful to the Referee for their positive report and we are pleased with their final recommendation.

---

## Round 2 · Referee Report · Anonymous (Referee 4) · 2024-7-5

Strengths

1. Novel approach for studying vortex dynamics in nontrivial geometries.
2. Interesting demonstration of method both inside and outside an ellipse.

Report

The updated manuscript has been significantly expanded, and now presents a much more general methodology for determining vortex dynamics and energetics in complex geometries. As such, I think the revised manuscript does open pathways for new resarch, and therefore satisfies the criteria for publication in SciPost Physics.

I have a few minor comments regarding the revisions, but once these have been addressed, I recommend the manuscript be accepted for publication.

Requested changes

The authors have addressed almost all my previous comments satsifactorily, although I have responses to some minor aspects of the first two points:
- The Jacobi theta function may be straightforward to calculate numerically, but mathematically it requires an infinite series to be defined, whereas sin(x) can be defined in terms of elementary operations (eg. as a ratio of two sides of a triangle, or via complex exponentiation). Since the theta function involves an infinite number of operations, I do not think it can strictly be referred to as "closed-form". Regardless, I leave the terminology up to the authors.

- A number of improvements have been made to the wording around the fluid compressibility in the introduction, but the second sentence still refers to superfluids having "negligible compressibility", which is not true for ultracold atomic gases (as I pointed out in my previous report). I suggest removing this.
* * *
I also have a few minor comments regarding the new material:
- Could the title be made more general? The emphasis of the revised manuscript seems to be more on the general method, rather than on the specific case of an elliptical geometry, but this is not reflected in the title.

- Paragraph 2 of the introduction reads: "Vortices in two-dimensional films have simpler dynamics...". This seems to be implying that the dynamics are simpler than in three-dimensional systems, but the preceding text does not mention three-dimensional systems specifically, so the word "simpler" appears out of place. I suggest altering this wording slightly.

- The sentence directly after this reads: "Hence a point-vortex model applies". This does not immediately follow from the dynamics being restricted to 2D. A point-vortex approximation also requires core sizes to be small compared to other relevant lengthscales (eg. the system size, distances to other vortices). I think this is worth mentioning here.

- Paragraph 4 begins: "Most cold atom experimental platforms are able to produce essentially uniform systems". I am not sure if "most" is true, so I suggest weakening this claim to eg. "many".

- The acronym "BEC" is defined twice in the introduction.

- In paragraph 8 of the introduction, the authors refer to "the vortex interpretation of the boundary-value problem". I am not sure what this means. Can the authors clarify this sentence?

- Regarding Eq (10), it would be helpful to specify that * denotes complex conjugation.

- I was confused by the definitions of $n_0$ and $n_i$, introduced in Sec 3.1:
i) Following Eq (20), $n_0$ is described as the "quantized circulation around the boundary", although it is not clear exactly which boundary is being referred to. Shortly afterwards, following Eq (23), $n_i = n_0 - 1$ is described as "the circulation quantum around the circular boundary at R". So are $n_i$ and $n_0$ defined around different boundaries? My best guess is that $n_0$ is the winding around the w-plane (elliptical) boundary, while $n_i$ is the winding around the z-plane (circular) boundary, but this appears to be reversed in the caption of Fig 1. I suggest the authors clarify this.
ii) Is the difference of -1 between the two winding numbers $n_i$ and $n_0$ related to the pole at the origin of the z-plane? It would be useful to specify where this difference comes from.

- Following Eq (29), the authors point out that for small distances $d \ll a$ from the ellipse, the energy diverges as $\log(2d)$. Is this easy to see from Eq (29)? Which terms contribute or drop out in this limit?

- It would be useful to include color bars in the figures with color scale data, or at least a description of the color scale in the caption (eg. for Fig 2, blue $\rightarrow$ low energy, orange $\rightarrow$ high energy).

Recommendation

Ask for minor revision

  • validity: top
  • significance: high
  • originality: high
  • clarity: high
  • formatting: excellent
  • grammar: excellent

Author:  Matteo Caldara  on 2024-07-18  [id 4630]

(in reply to Report 2 on 2024-07-05)
Category:
remark
answer to question

We would like to thank the Referee for their positive and constructive report. We are glad our revised manuscript is considered worth publication in SciPost Physics. We also thank the Referee for their new suggestions which we implemented in the new submitted version of our manuscript.

In the attached pdf document we provide a a complete point-by-point reply, followed by a "diff" file, highlighting the latest changes to our manuscript.

Attachment:

Reply_and_Diff.pdf

---

## Round 2 · Author Response

Dear Editor,

Thank you for sending us the reports of the two Referees.
At the moment, we would decline the transfer to the journal SciPost Physics Core. Instead, we would like our resubmission to be considered for publication in the journal SciPost Physics.
In light of Referees' comments and requests, we have substantially modified our manuscript in order to highlight the original contributions that our work can bring to the field of superfluid vortex dynamics, as well as how it now fulfills the journal acceptance criteria. We summarised the general structure of the new version of our paper at the beginning of the reply to both Referees.

We have replaced the arXiv manuscript with the revised version that seeks to comply with the requests of both Referees. Our detailed point-by-point response to the Referees has been uploaded, too. In our response, we also explicitly reference the acceptance criteria.

In view of the considerable improvements that we have implemented in response to the points raised by the Referees, we are confident that this revised version of our paper can be accepted for publication in SciPost Physics.

Best regards,
The authors

---

## Round 2 · List of Changes

A detailed list of changes can be found at the beginning of the reply to both Referees that has been attached as a pdf file.

---

## Round 3 · Author Response

Dear Editor,

Thank you for sending us the reports of the two Referees. We have addressed all the points raised by the second Referee, implementing their requests in the revised version of our paper. We have updated our arXiv manuscript and we have uploaded our response to the Referees.

In view of these latest improvements, we are confident that our work can be ready for publication in SciPost Physics.

Best regards,
The authors

---

## Round 3 · List of Changes

A detailed list of changes has been attached as a pdf file in the response to the second Referee.

---

## Editorial Decision

published